# Depolarization Characteristics of Different Reflective Interfaces Indicated by Indices of Polarimetric Purity (IPPs)

**DOI:** 10.3390/s21041221

**Published:** 2021-02-09

**Authors:** Dekui Li, Kai Guo, Yongxuan Sun, Xiang Bi, Jun Gao, Zhongyi Guo

**Affiliations:** School of Computer and Information, Hefei University of Technology, Hefei 230009, China; 2019170964@mail.hfut.edu.cn (D.L.); kai.guo@hfut.edu.cn (K.G.); syx@hfut.edu.cn (Y.S.); bixiang@hfut.edu.cn (X.B.); gaojun@hfut.edu.cn (J.G.)

**Keywords:** samples, incident angles, roughnesses, IPPs

## Abstract

Compared with the standard depolarization index, indices of polarimetric purity (IPPs) have better performances to describe depolarization characteristics of targets with different roughnesses of interfaces under different incident angles, which allow us a further analysis of the depolarizing properties of samples. Here, we use IPPs obtained from different reflective interfaces as a criterion of depolarization property to characterize and classify targets covered by organic paint layers with different roughness. We select point-light source as radiation source with wavelength as 632.8 nm, and four samples, including Cu, Au, Al and Al_2_O_3_, covered by an organic paint layer with refractive index of n = 1.46 and Gaussian roughness of α = 0.05~0.25. Under different incident angles, the values of *P*_1_, *P*_2_, *P*_3_ at divided 90 × 360 grid points and their mean values in upper hemisphere have been obtained and discussed in the IPPs space. The results show that the depolarization performances of the different reflective interfaces (materials, incident angles and surface roughness) are unique in IPPs space, providing us with a new avenue to analyze and characterize different targets.

## 1. Introduction

Reflecting properties of different surfaces can be applied in many areas, such as 3D graphics simulation [1], biological sensing [2,3] and prediction of vegetation [4,5], which has been investigated extensively in recent years. Polarization as a powerful tool plays a great role in Mie ellipsometry [6,7] or full radiative transfer simulations [8]. In addition, polarization information of reflected light, such as degree of polarization (DoP), angle of polarization (AoP), Stokes vector et al., can provide more additional information about the targets in new aspects. When light interacts with targets, the polarization of the incident light may undergo a modification, carrying structural information of the targets. Therefore, it is important to characterize the process of the reflective polarization for exploring the polarization distributions and revealing the features of the surface, which can be used in polarization remote sensing [9,10], target detection and recognition [11,12,13,14], especially in the hidden-target identification. When the sample is not an ideal smooth surface, the reflected light will have different propagating directions, namely scattering. Reflection from a rough surface can be typically modeled using a bidirectional reflectance distribution function (BRDF) [15,16,17,18,19,20,21,22,23]. As a famous geometrical optics model, the Torrance–Sparrow (T-S) model [15] has been widely investigated and expanded to the polarized BRDF (pBRDF) models [16,17,18,19,20,21]. The Monte Carlo (MC) method has also been introduced to solve the composition of samples in fusion devices or reactive plasmas and a series of cases about polarization information [24,25,26,27,28,29,30,31,32,33], especially in photons tracking [24,25,29,30]. Wang et al. have combined the MC model with BRDF, giving birth to a flexible method to acquire the reflective polarization information from a rough surface [34]. However, these models cannot represent depolarization of the samples, which is important in real applications [35,36,37,38,39,40,41,42].

As mentioned above, depolarization is related to the structural and chemical characteristics of the samples at different physical scales, and it can be used for classifying [35,36] and analyzing samples [38,39,40,41,42]. The Mueller matrix (MM), which is composed of 4 × 4 elements namely m_ij_, can be used to characterize the depolarization properties of mediums, and from the obtained depolarization we can obtain a set of parameters, namely IPPs (indices of polarimetric purity). The IPPs are composed of three mutually orthogonal axes *P*_1_, *P*_2_, and *P*_3_, providing a detailed description of the polarimetric purity [43,44,45]. In theory, the obtained IPPs can form a purity space. Each point in the purity space can represent a specific degree of depolarization very well, which is linked to the relative weights of the spectral components of MM directly. Meanwhile, it also provides complete information on the structure of polarimetric randomness. Given an MM, the corresponding values of *P*_1_, *P*_2_, *P*_3_ can be directly calculated [43,44,45,46,47]. In simple words, IPPs are easy to implement for discussing depolarization properties.

In this paper, we investigated the depolarization of the reflective interfaces that are composed of Cu, Au, Al and Al_2_O_3_ covered by an organic paint layer, which is a kind of common paint with a refractive index of n = 1.46 under the wavelength as 632.8 nm, and different roughness as functions of incident angles by using MC simulations. The overall depolarizations of different samples are calculated, and the results illustrate that the overall depolarization increases with increasing the roughness of organic paint layers. In addition, the metals and oxides samples show opposite dependence on incident angles, in which large incident angles may result in weak depolarization of metals, and on the contrary, result in strong depolarization of oxides. Moreover, these dependences are also varying for different metals. The exiting light from reflective interface has different IPPs with different zenithal and azimuthal directions. In other words, refractive light carries unique depolarization information with different directions. In order to better explain the depolarization characteristics of samples, we also calculate *P*_1_, *P*_2_, *P*_3_ at each divided grid point in the upper hemisphere space that include all directions of refractive light. Our results can reflect more dimensions information about depolarization of the target, which provides a new method for remote sensing and hidden target identification.

## 2. Methods

### 2.1. The IPPs of Material Media

To start, we focused on a set of three parameters, called IPPs, which provide an extended way to characterize, analyze and classify samples with different depolarization property. It is because IPPs can represent the relative statistical weight of the decomposed nondepolarizing components, providing a more accurate description of the depolarization properties of the sample. IPPs are defined as relative differences among the four eigenvalues (taken in decreasing order $\lambda_{0}\geq \lambda_{1}\geq \lambda_{2}\geq \lambda_{3}\geq 0$) of the covariance matrix **H,** which can be obtained from MM by the following equation [43]:(1)$$H=\frac{1}{4}(\begin{array}{ccc} m_{00}+m_{01}+m_{10}+m_{11} & m_{02}+m_{12}+i\left(m_{03}+m_{13}\right) & m_{20}+m_{21}-i\left(m_{30}+m_{31}\right)\\ m_{02}+m_{12}-i\left(m_{03}+m_{13}\right) & m_{00}-m_{01}+m_{10}-m_{11} & m_{22}-m_{33}-i\left(m_{23}+m_{32}\right)\\ m_{20}+m_{21}+i\left(m_{30}+m_{31}\right) & m_{22}-m_{33}+i\left(m_{23}+m_{32}\right) & m_{00}+m_{01}-m_{10}-m_{11}\\ m_{22}+m_{33}-i\left(m_{23}-m_{32}\right) & m_{20}-m_{21}+i\left(m_{30}-m_{31}\right) & m_{02}-m_{12}-i\left(m_{03}-m_{13}\right) \end{array}\;\;\begin{array}{l} m_{22}+m_{33}+i\left(m_{23}-m_{32}\right)\\ m_{20}-m_{21}-i\left(m_{30}-m_{31}\right)\\ m_{02}-m_{12}+i\left(m_{03}-m_{13}\right)\\ m_{00}-m_{01}-m_{10}+m_{11} \end{array})$$

In theory, four non-negative eigenvalues ($\lambda_{0}\geq \lambda_{1}\geq \lambda_{2}\geq \lambda_{3}\geq 0$) can be obtained from the covariance matrix **H**, which can be used to represent the relative statistical weights of the nondepolarizing components, from which the IPPs can be defined by the following equations:(2)$$P_{1}=\frac{\lambda_{0}-\lambda_{1}}{\mathrm{tr}H},$$
(3)$$P_{2}=\frac{\left(\lambda_{0}-\lambda_{2}\right)+\left(\lambda_{1}-\lambda_{2}\right)}{\mathrm{tr}H},$$
(4)$$P_{3}=\frac{\left(\lambda_{0}-\lambda_{3}\right)+\left(\lambda_{1}-\lambda_{3}\right)+\left(\lambda_{2}-\lambda_{3}\right)}{\mathrm{tr}H},$$

The depolarization index ($P_{\Delta }$) can be calculated from IPPs as follows:(5)$$P_{\Delta }=\frac{1}{\sqrt{3}}\sqrt{2P_{1}^{2}+\frac{2}{3}P_{2}^{2}+\frac{1}{3}P_{3}^{2}},$$

In general, nondepolarizing samples are characterized by $P_{\Delta }$ = *P*_1_ = *P*_2_ = *P*_3_ = 1. The samples with $P_{\Delta }$ = *P*_1_ = *P*_2_ = *P*_3_ = 0 are corresponding to ideal depolarizers, and there will be $\mathrm{MM}\;=\;diag\left(m_{00},\;0,\;0,\;0\right)$, where $m_{00}$ is the mean intensity coefficient. Note that, the values of IPPs will be restricted by the following inequalities:(6)$$0\leq P_{1}\leq P_{2}\leq P_{3}\leq 1,$$

### 2.2. Microfacet Theory

In general, the height field of rough surface satisfies the Gaussian distribution with variance α. The larger the variance, the rougher the surface. In addition, rough surface is assumed to be made of many microfacets, called microfacets theory, and the normal vector of each microfacet can be determined by θ and σ shown in Figure 1b, which can be calculated by sampling [34]. Then, the reflective light and refractive light of microfacets can be calculated by the Fresnel formula and normal vector of microfacet [48,49].

In the tracking process of the polarized light, each beam of light propagates in global coordinate system shown in Figure 1a, and Fresnel’s law is used in the local coordinate system defined by normal vector of microfacets shown in Figure 1b. Thus, it is necessary to translate two kinds of coordinate system. The coordinate transformation can be accessed by rotating an angle θrot [48]. The rotation matrix is $R{\left(\theta \right)}_{rot}$, and the corresponding Mueller matrix $M{\left(\theta \right)}_{rot}$ can be expressed as follows [48].
(7)$$R{\left(\theta \right)}_{rot}=\left(\begin{array}{cc} \cos\theta & -\sin\theta \\ \sin\theta & \cos\theta \end{array}\right),$$
(8)$$M{\left(\theta \right)}_{rot}=\left(\begin{array}{cccc} 1 & 0 & 0 & 0\\ 0 & \cos2\theta & -\sin2\theta & 0\\ 0 & \sin2\theta & \cos2\theta & 0\\ 0 & 0 & 0 & 1 \end{array}\right),$$

### 2.3. Monte Carlo Simulation

As shown in Figure 2a, a sample is covered by some coating layers. The point-light source is emitted in the upper hemisphere. When a beam of light reaches to the coating layer with an angle of θ, refraction and reflection will happen. The reflected light will be collected in the upper hemisphere immediately. In contrast, the refracted light will go through a series of reflections and refractions, and can be collected in the upper hemisphere eventually, in which the upper hemisphere is divided into 90 × 360 grids with the step of 1° (in both zenithal and azimuthal directions) to collect the reflective and refractive photons. The top view of the upper hemisphere is shown in Figure 2b, in which the upper hemisphere can be divided into 90 rings according to the zenith angle (0°~90°), and combining azimuth angle (0°~360°), the detection grid can be fixed. The exiting light from the coating layers can be collected according to their concrete positions defined by the azimuth and zenith in the corresponding grid, from which their Stokes vectors in each grid can be obtained by counting and summing the received photons’ Stokes vectors.

The MC method is adopted to perform the simulation [34]. In order to explore the depolarization of samples covered by organic layers, we needed to get the MM of each grid under conditions of different incident angles and roughnesses of organic layers. Thus, we have defined the direction of the incident light as 30°, 40°, 50°, 60° and 70°, four kinds of incident lights with different polarization states as $S_{in1}$= [1000], $S_{in2}$= [1100], $S_{in3}$= [1010], $S_{in4}$= [1001], respectively, the samples under organic layers as Cu, Al, Au, and Al_2_O_3_, and the roughnesses of the organic layers as 0.05~0.25 with the step of 0.05. Based on experience, we have fixed the roughness of sample as 0.2. In the simulation model, light reaching the coating layer can be traced as the following steps:

Calculating the next layer j (usually i +1 or i −1) to be scattered based on the number of current layer i and the direction of propagation.Sampling the normal vector of the $j_{th}$ layer according $\alpha_{x}$ and $\alpha_{y}$ [34].Transforming polarized light S from the global coordinate to the local coordinate by $S_{l}=M{\left(\theta \right)}_{rot}S$.Calculating the direction of reflected and refracted light according to the Fresnel formula and normal vector on the selected microfacet.Obtaining reflected and refracted light from Fresnel’s formula and MM, respectively, by $S_{rl}=M_{r}S_{l}$, $S_{tl}=M_{t}S_{l}$, where $M_{r}$ and $M_{t}$ are the reflective Muller matrix and the transmitting Muller matrix, respectively [34].Translating them from the local coordinate to the global coordinate, respectively, by $S_{r}=M{\left(\theta \right)}_{rot}S_{rl}$,$S_{t}=M{\left(\theta \right)}_{rot}S_{tl}$.Checking whether the light has left the coating. If yes, collecting the light in the upper hemisphere. If no, back to step 1.Calculating MM and covariance matrix of each grid in the upper hemisphere.Getting the eigenvalue $\left(\lambda_{0}\geq \lambda_{1}\geq \lambda_{2}\geq \lambda_{3}\right)$ from the covariance matrix.Calculating *P*_1_, *P*_2_, *P*_3_ from $\lambda_{0}\lambda_{1}\lambda_{2}\lambda_{3}$.

The process of photons tracking by MC is shown in Figure 3.

It should be noted that our simulation was based on two assumptions: (1) the scale of microfacet is much larger than the incident wavelength, which means the geometry optics can be applied; (2) the coating is so thin that the absorption can be ignored.

## 3. Results and Discussion

### 3.1. Comparing with BRDF Model

To demonstrate the accuracy and validity of the simulation model, we compared the results of reflection from a copper surface obtained by MC simulation and experiment-based BRDF model included in the SCATMECH [50]. Here, it should be noted that the SCATMECH is a light scattering library and published by the NIST in 2017 [50], which has been verified in many experiments. In both simulation schemes, the refractive index of copper and the surface roughness parameters are set as 0.27 + 3.40*i* and α_x_ = α_y_ = 0.2, respectively, and the incident nonpolarized light (with the wavelength of 632.8nm and the incident angle of 40°) is set as *S**in* = [1000] with 10 million emitted photons every time. The results of our MC model and the experiment-based BRDF model are plotted in the top and bottom panels in Figure 4, respectively. It is obvious that the reflective polarization patterns of I, Q, U, V, AoP and DoP obtained by our MC model agree well with those obtained by analytical BRDF model, which can verify the accuracy and validity of our model.

### 3.2. Influence of Roughness

First, we chose samples (Cu, Al, Al_2_O_3_, Au) covered by organic paint layers as different reflective interfaces, in which the organic paint layer is a common paint and we assume it is a pure substance whose refractive index is 1.46 under the incident wavelength of 632.8nm. We would investigate the dependence of *P*_1_, *P*_2_, and *P*_3_ on the roughness of the organic paint layer. In the simulation, the roughness α_x_ = α_y_ ranges from 0.05 to 0.25 with step of 0.05. The incident angle and wavelength of light are set as 632.8 nm and 50° in the upper hemisphere, respectively. We have investigated four samples, including Cu (*n* = 0.27 + 3.40i), Al (*n* = 1.4482 + 7.53i), Au (*n* = 0.18 + 3.43i), and Al_2_O_3_ (*n* = 1.77), which are covered by an organic paint layer. The number of emitted photons is 10 million to ensure the accuracy of our MC simulation.

To study the overall depolarization property, we calculated the average values of *P*_1_, *P*_2_, and *P*_3_ at all physically feasible points [46] in the upper hemisphere of reflective interface, as shown in Figure 5. It can be observed that *P*_1_, *P*_2_, and *P*_3_ form IPPs space, in which the point (1, 1, 1) represents nondepolarizing samples and the other points represent depolarizing samples. In other words, the intrinsic depolarizing mechanisms can be demonstrated according to the coordinate in the IPPs space. Figure 5 shows that the values of *P*_1_, *P*_2_, and *P*_3_ gradually decrease with increasing roughness, indicating the depolarization of the samples increases with the increasing roughness. It is because light will be scattered rather than reflected at a rough surface, leading to depolarization of light. In addition, we can see that the calculated results of Cu in IPPs space are closer to that of Au, while far away from those of Al and Al_2_O_3_. This phenomenon could be attributed to the refractive index of the samples. As shown above, the refractive index of Cu is similar to that of Au, but quite different from those of Al and Al_2_O_3_. Therefore, we may get different distributions in the IPPs space for different samples, making it possible to classify the samples.

In order to further demonstrate the advantages of IPPs, we calculated the values of *P*_1_, *P*_2_, and *P*_3_ at each grid in the upper hemisphere when the surface roughness of organic paint layer is α_x_ = α_y_ = 0.05, 0.10, 0.15, 0.20, and 0.25, as shown in Figure 6. Here, we take the sample of Cu as an example. The points at which the values of *P*_1_, *P*_2_, and *P*_3_ equal zero means that eigenvalues derived from the covariance matrix **H** are negative, called physically unfeasible points [46]. The number of these feasible points increases but the values are decreasing with increasing roughness, which means the average value is decreasing for all physically feasible points when the roughness increases. It is consistent with Figure 5. From the simulation results, we can obtain particular depolarization information of the sample from the physically feasible points. For example, the values of *P*_1_, *P*_2_, and *P*_3_ in the point of (60, 0) decrease with increasing roughness, but are always bigger than those in the point of (60, 60). It means that the MM for the latter case has more depolarization components in characteristic decomposition, which reflects the different points in the upper hemisphere having different depolarization components. In other words, light received at different points in the upper hemisphere undergoes various coding by the sample. This characteristic makes it more difficult for us to classify the samples by using the values of *P*_1_, *P*_2_, and *P*_3_ at each grid in the upper hemisphere than by using their average values’ distributions in IPPs space.

### 3.3. Influence of Incident Angle

It is well known that the scattering of light at an irregular structure is highly dependent on the incident angle. Therefore, the dependences of depolarization of samples on the incident angles were investigated. Here, we chose the same reflective interface, but the surface roughness of the organic paint layer was fixed as α_x_ = α_y_ = 0.2. The incident angles are 30°, 40°, 50°, 60° and 70°. The calculated overall distributions of *P*_1_, *P*_2_, and *P*_3_ in IPPs space are shown in Figure 7.

For metals, the results show that their depolarization properties decrease with the increasing incident angles. It is because light collected at most grids has smaller scattering components with increasing incident angles. On the contrary, oxides, such as Al_2_O_3_, hold opposite results that increasing incident angles result in more depolarization. These results illustrate that metals and oxides have different dependence of depolarization characteristics on incident angles, which makes it possible for us to classify samples.

Similarly, the values of *P*_1_, *P*_2_, and *P*_3_ at each grid in the upper hemisphere is not significantly dependent on the incident angles, as shown in Figure 8. Here, we still take the sample of Cu as an example. It can be seen that the number of physically unfeasible points slightly decreases with the increasing incident angles, and the physically feasible points have a tendency of spreading towards the center of the circle under a large incident angle. The depolarization performances at different grids are different. In addition, the *P*_1_, *P*_2_, *P*_3_ of the same grid are different, which is because the *P*_1_, *P*_2_, *P*_3_ as the relative differences of different pure systems mapped from the reflective interface depend on the inherent attribute of reflective interface, which can be used for analyzing IPPs decomposition of reflective interface and exploring the composition of reflective interface. Combining the distribution patterns of *P*_1_, *P*_2_, *P*_3_ and the IPPs space has significant advantages in classifying the depolarization characteristics of samples.

## 4. Conclusions

In this paper, we have emphasized the interest of using IPPs as a criterion for characterization and classification of samples covered by organic paint layers. On one hand, the IPPs carry unique depolarization information of samples, thus leading the unique distributions of overall depolarization for different samples in IPPs space. The distributions of Cu, Al, Au, Al_2_O_3_ with different incident angles and roughnesses of organic paint layers were investigated and discussed. On the other hand, the IPPs of each grid vary, which represents that the light coded by samples vary in different directions. These have exhibited the significant potential of using IPPs for target detection and remote sensing, especially the identification of the hidden target.

## Figures and Tables

**Figure 1 sensors-21-01221-f001:**
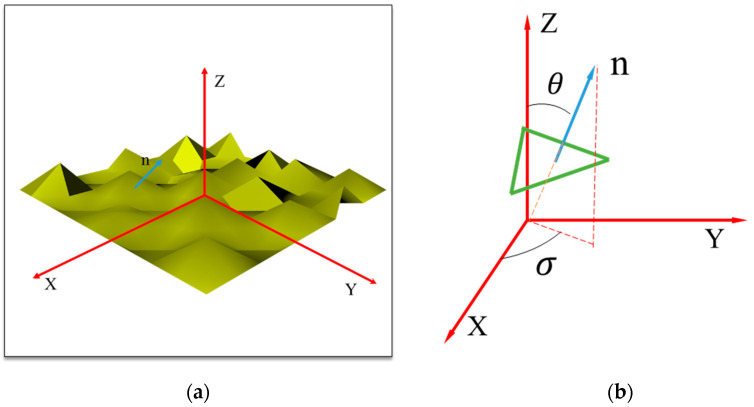
(**a**) Surface composed of the microfacets in the XYZ coordinate system, (**b**) the schematics of a single microfacet.

**Figure 2 sensors-21-01221-f002:**
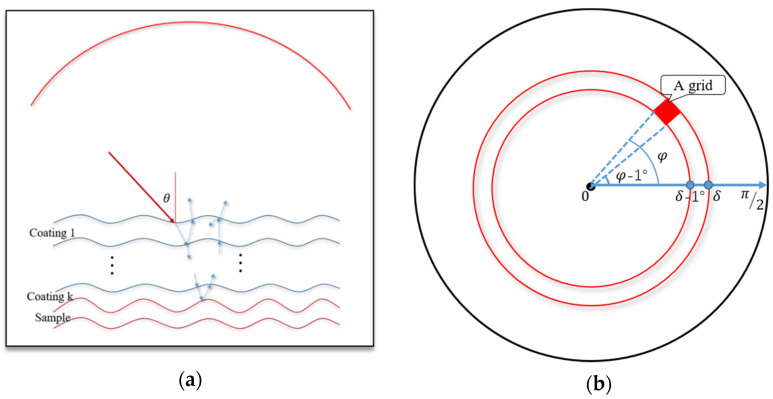
(**a**). Model of sample covered by coating layers, (**b**) the top view of the upper hemisphere.

**Figure 3 sensors-21-01221-f003:**
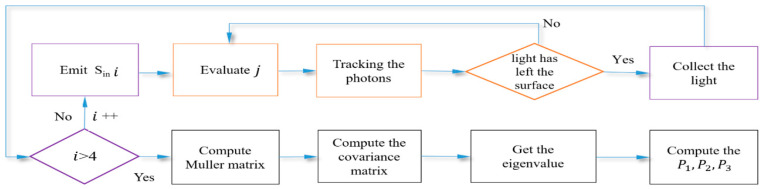
The flow chart of Monte Carlo.

**Figure 4 sensors-21-01221-f004:**
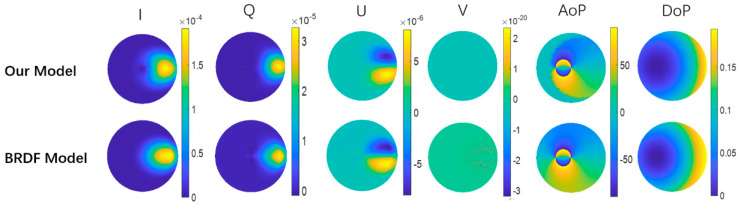
Comparison of the results obtained by our model and the analytical bidirectional reflectance distribution function (BRDF) model.

**Figure 5 sensors-21-01221-f005:**
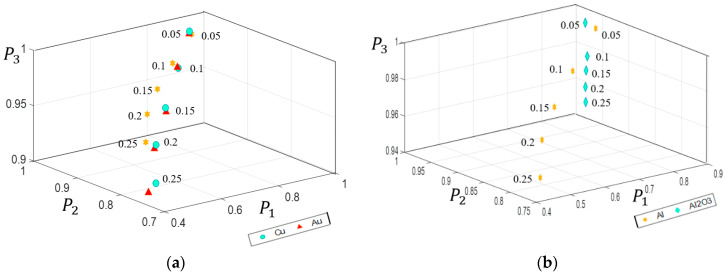
The overall distributions of *P*_1_, *P*_2_, *P*_3_ corresponding to different roughnesses (**a**) samples as Cu, Au, Al; (**b**) samples as Al, Al_2_O_3._

**Figure 6 sensors-21-01221-f006:**
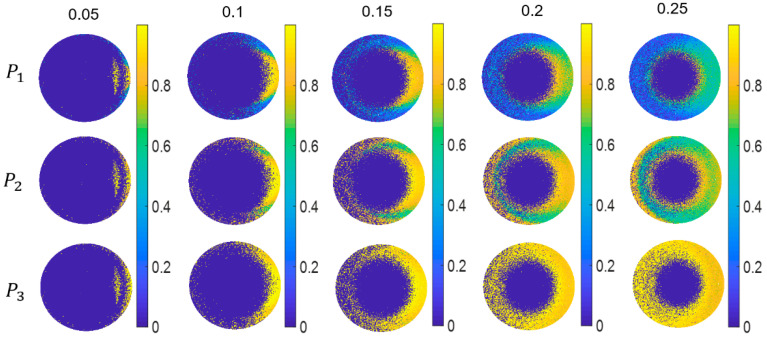
*P*_1_, *P*_2_, *P*_3_ of each grid in the upper hemisphere with the changing roughness of Cu targets.

**Figure 7 sensors-21-01221-f007:**
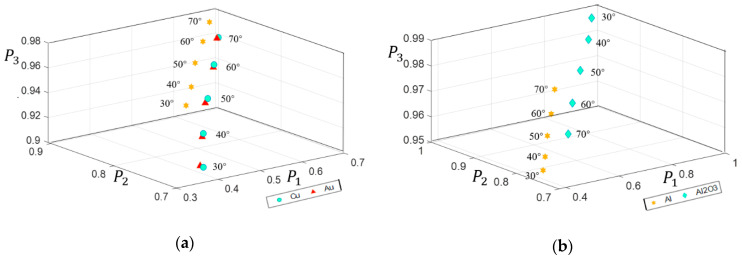
The overall distributions of *P*_1_, *P*_2_, *P*_3_ corresponding to different incident angles: (**a**) samples as Cu, Au, Al; (**b**) samples as Al, Al_2_O_3._

**Figure 8 sensors-21-01221-f008:**
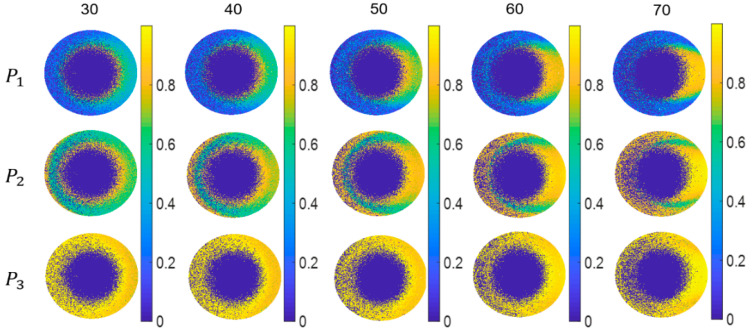
*P*_1_, *P*_2_, *P*_3_ of each grid in the upper hemisphere with the incident angle change for the Cu target.

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
