# Peer review of "Depolarization Characteristics of Different Reflective Interfaces Indicated by Indices of Polarimetric Purity (IPPs)"

_sensors, 2021, doi:10.3390/s21041221_

Round 1
Reviewer 1 Report
The referee report is given in the attached document.

Author Response
See the attached files.

Reviewer 2 Report
The authors present a numerical method of characterizing samples of some materials by means of the indices of polarimetric purity (IPP). The reflected photons are supposed to be received in a hemispheric array of detectors. Results are consistent by themselves and the test of the scheme, put in comparison with that obtained with some well-known public routines (from Scatmech) is satisfactory. The IPP set is obtained from the received photons, emitted with different angles of incidence over the samples, assuming different roughness values in the samples and for different substrates.
The most apparent weak point of the work, in my opinion, is the explanation about the structure of the samples. They speak about some organic material put over a metallic plate, but it is difficult to get a clear idea of what is the generic structure they are analyzing, or how many discontinuities or layers there are in it. On the other hand, reader cannot appreciate if this kind of samples is specifically connected with some real problem, i.e., the optical analysis of a tissue species. I suppose they use a metallic plate only to collect a higher number of reflected photons. If this is the point, they should say so openly.
Author Response
See the attached files.

Round 2
Reviewer 1 Report
I thank the authors for considering many of my previous comments and for the very comprehensive reply. I suggest to add a few further points to the manuscript before publication.
1. Line 100: “The rotation matrix of the three-dimensional coordinates is R(Theta)_rot … “.
R(Theta)_rot is, however, a 2x2 matrix, and can’t be a rotation matrix for 3D.
2. Line 114: “... from which their Stokes vectors in each grid can be obtained by counting and summing the received photons’ degree of polarizations.”
Isn't it the other way? The photons of the Stokes vector are counted and summed up, and afterwards the final polarization is determined from the Stokes vector at this position?
3. Line 133. What are M_r and M_t?
4. Caption of Fig. 6: Could you mention here the target material?
As all of the mentioned points should be straightforward, it is not necessary to have a further review iteration. I therefore recommend for publication after these changes are made.
With kind regards,
Anonymous referee
